# Modeling of Dental Implant Osseointegration Progress by Three-Dimensional Finite Element Method

**Iulia Roatesi [1] and Simona Roatesi [2,\*]** 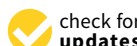

[1]  Department of Histology and Cytology, Dental Medicine Faculty, *Carol Davila* University of Medicine and Pharmacy, 050474 Bucharest, Romania; iulia.roatesi@umfcd.ro

[2]  Department of Applied Informatics, *Ferdinand I* Military Technical Academy, 050141 Bucharest, Romania

\*  Correspondence: simona.roatesi@yahoo.fr or simona.roatesi@mta.ro; Tel.: +40-723-288-108

**Abstract:** As osseointegration is a time-dependent process, biomechanical assessment is thought to determine whether a fibrous encapsulation or a bone covering will develop around an implant, according to the stress in the implant and surrounding bone. This study proposes a model for stress evaluation by finite element method (FEM) during the osseointegration progress, the main factor implied in implant success or failure. The loadings due to masticatory forces generate stress concentration and consequently, an adequate risk concerning the implant stability should be assessed. An accurate FEM model is used to calculate the stress and displacement in the whole implant–bone system during the osseointegration progress. This process is simulated by taking into account the gradual increase in the damaged biomechanical properties of the cortical bone. The results reveal that as the implant osseointegration occurs gradually, the bone stiffness from the peri-implant area increases gradually, such that in the end (healing) we observed that the cortical bone begins to take over the bending loading. In addition, the displacements decrease as the osseointegration gradually occurs and the cortical bone stress reaches higher values, which are placed in the mandibular ridge. The FEM is suitable to model the osseointegration progress, offering valuable information concerning the stress concentration zones in the implant–bone system and consequently, the risk evaluation, both for pre- and post-osseointegration.

**Keywords:** dental implantation; dental crown; biomechanics; numerical modeling; dental implant stability

## 1. Introduction

Finite element method (FEM) is a helpful tool in the study of dental implantology as it allows the determination of the state of stress, strain and displacement in the implant and the surrounding bone [1–4]. A FEM analysis is very important in implants' optimal design [5–8], in assessing the factors that influence the whole process of pre- and post-osseointegration [9–11], or they are dedicated to oral rehabilitation [12,13], etc.

A very important issue in oral implantology is the study of the osseointegration process [14,15]. It is difficult to rigorously assess this process, whereas the biomechanical response differs from one patient to another [16,17] and moreover, the process itself depend on the specific factors that may relate to osseous healing around an implant.

Generally, FEM allows the study of the behavior of the implant and surrounding bone tissue after the osseointegration process termination [18]. In this context, it is considered a fixed contact between the bone tissue and implant throughout the bone–implant interface, i.e., under loading, there is no

relative motion between the bone and implant, which would model complete osseointegration. In this case, the analysis carried out corresponds to a completely osseointegrated implant, and so for a time at least six months after the insertion of the implant. There are FEM analyses focusing on the investigation of the interaction between the implant and peri-implant bone tissue [19] or on the biomechanical of marginal bone resorption around osseointegrated implants [20].

Instead, if we consider an analysis of stress and displacement state during a period in the first few weeks or months after the implant insertion, when osseointegration is not fully attained, we propose an original method of an incomplete osseointegration modeling.

The numerical simulation using FEM determines the stress and displacement state both immediately after the implant insertion, as well as during the consolidation process of the bone–implant interaction phenomena until complete healing. The model under consideration in our study is that of a premolar implant on the jaw with a temporary crown for aesthetic reasons, in a lighter contact.

Therefore, the aim of this paper is to study by FEM the osseointegration process modeling since the first weeks of installing the implant to the complete osseointegration.

## 2. Materials and Methods

As the geometric model of the structure which is made up of the dental implant, bone and crown needs special preprocessing resources, the Solid Works program [21] was used to realize the model. The geometric model produced with this program was exported and used for calculations by the Cosmos program.

This study was dedicated to the analysis of the process of insertion of a dental implant into a section of jaw, with particular emphasis on highlighting the estimation for the various stages of osseointegration of the stress concentration zone in the bone and implant components under mastication loads. These areas represent the most vulnerable areas, in which the eventual yielding, rupture, damage of the structure may occur.

### 2.1. The Geometric Model of the Dental Implantation

The model of our analysis was static and elastic materials were considered.

Both the geometric and the finite element (FE) models of implant components and bone tissue were conducted with high accuracy, taking into account all the constructive and functional details (connection radii, threads, release cutting, contacts) so that the model could be as close as possible to reality.

Considering the case of a system made up of an implant [4], Rootform type, with a length of 11.5 mm, with a maximum thickness of 3.8 mm with two threaded areas (fine and large pitch) inserted in a portion of jaw extended for about 20 mm from the implant axis, as shown in Figure 1a,b.

The geometric model was made on one hand of the biological material, the jaw, composed of trabecular bone and cortical bone, and on the other hand, the implant, abutment and crown (Figure 1b). All these components were created by computer and used in the calculation by FEM. The implant was considered to be made of titanium alloy, the abutment of magnesium alloy and the crown of ceramic.

The implant is cone-shaped with two threaded zones (Figure 2).

The interior of the implant allowed the insertion with no thread of the intermediate part (abutment) (Figure 2b). At the top, it had a hexagonal reaming used to mount it in the jaw using an Allen key. The inner screw served to assemble the implant with the abutment by a titanium M2 screw (Figure 2b).

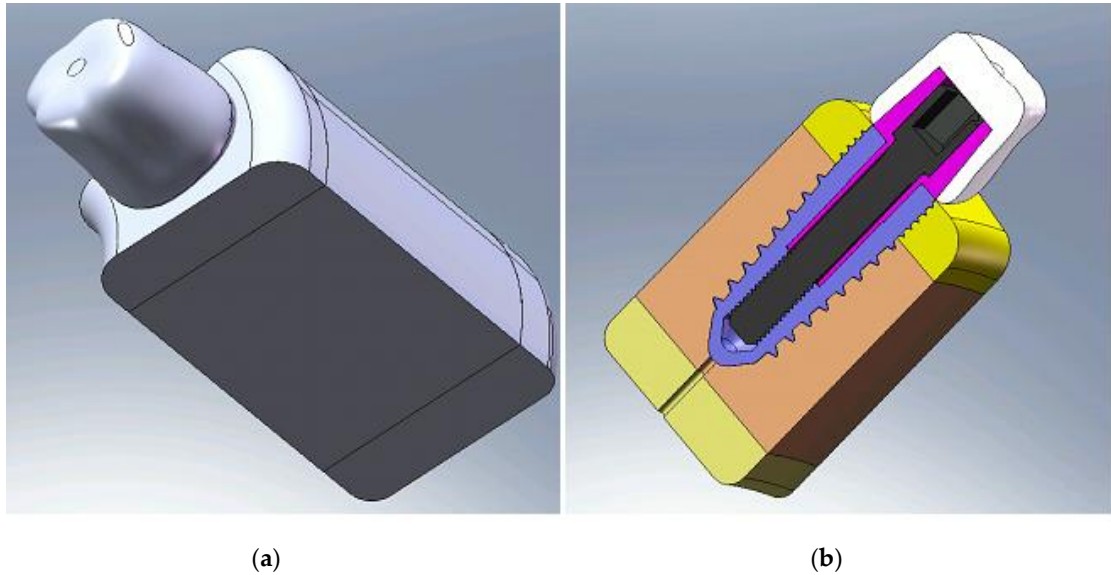

**Figure 1.** Geometrical model of the crown–implant–bone structure: (**a**) structure overview; (**b**) structure section.

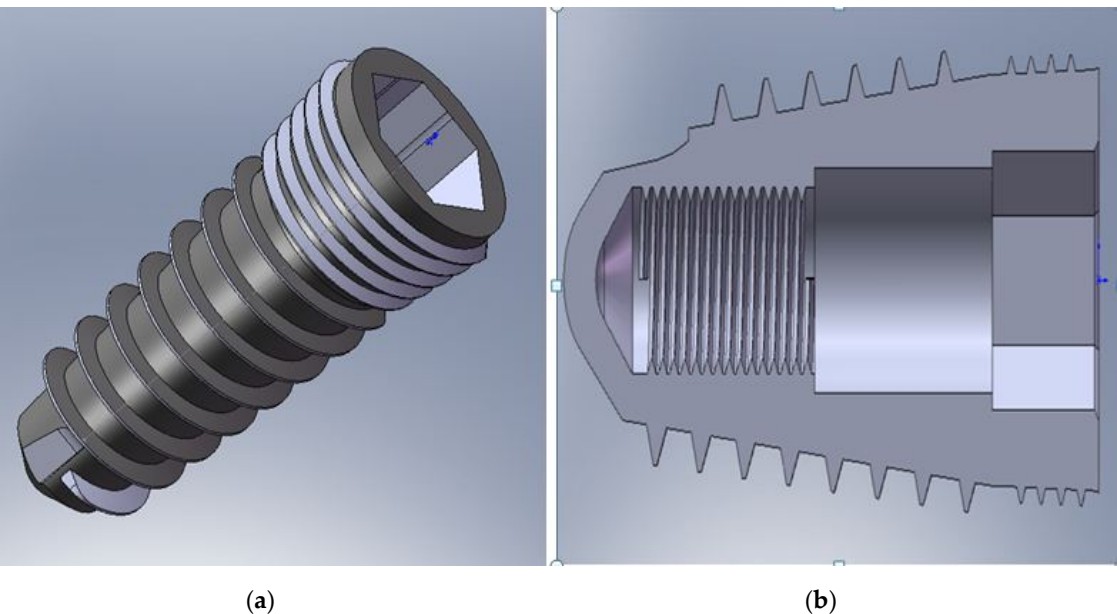

**Figure 2.** Geometrical implant model: (**a**) implant overview; (**b**) implant longitudinal section.

The crown geometrical model was shaped as close as to the actual shape (Figure 3a), being prepared to take into account in detail the crown characteristics, to avoid the biting force application on an almost flat surface that could affect the force transmission. The crown material was ceramic, even it was considered a provisional one. If there is no question of cost, and given that the mechanical properties of ceramics are clearly superior, this material is not outworn over time, referring to the masticatory surface (the resin from which the temporary crown is usually made easily becomes abrasive and in a short period of time), this solution can be adopted. Moreover, by using ceramics, which provide the brightness and translucency of the natural tooth, the aesthetic part is clearly superior, and if the patient, even if for a short period has maximum aesthetic requirements, it can use this solution. The supporting bone is represented by a portion of jaw around the implant on a certain distance to a boundary to which it is considered that there is no longer the influence of implant surgery.

Jaw modeling takes into account the different structure of the bone tissue (trabecular and cortical) (Figure 3b) by designating the respective zones to the corresponding material properties.

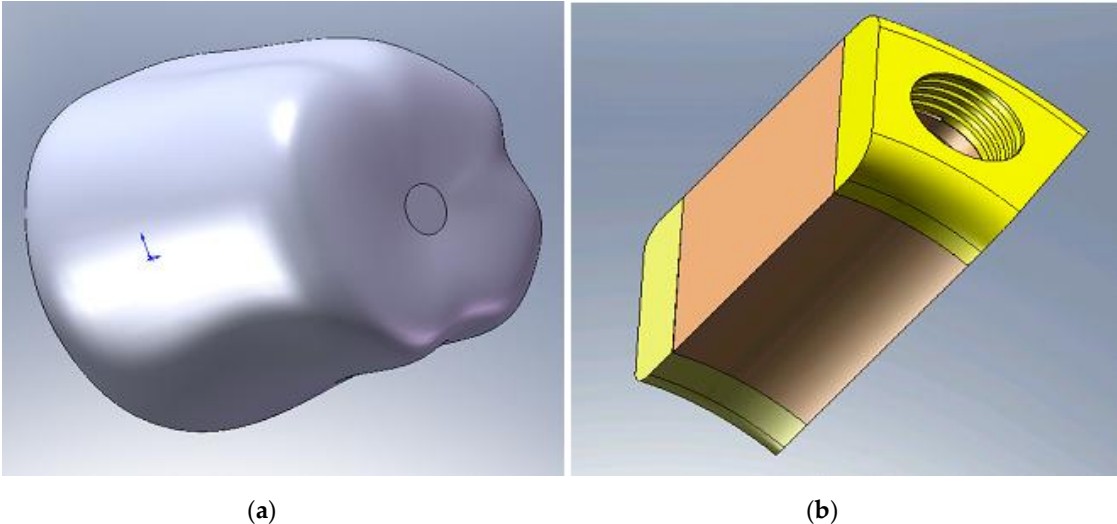

| (**a**) | (**b**) |

**Figure 3.** (**a**) Crown; (**b**) bone tissue with the layers of cortical bone (yellow) and trabecular bone (brown).

All components were modeled respecting in detail all the features of the actual model (threads, tapers, undercut, neck, etc.). The system is designed in such a way to transmit the masticatory force from the crown to the intermediate component and then to the upper annular part of the implant. During the transmission mechanism of mastication, force does not intervene in any threaded assembling.

## 2.2. Finite Element Model of the Dental Implantation

The 3D model used to study an implant in a portion of the jaw was built with the SolidWorks program and used tetraedrale elements in the implant and in the bone tissue as well [21]. The following shows one option of mesh which permits easy the observation of details under consideration. The finite element model consists of the corresponding parts of implant components, as shown in Figure 4a and the supporting bone. Figure 4b represents the finite element (FE) model of the whole system.

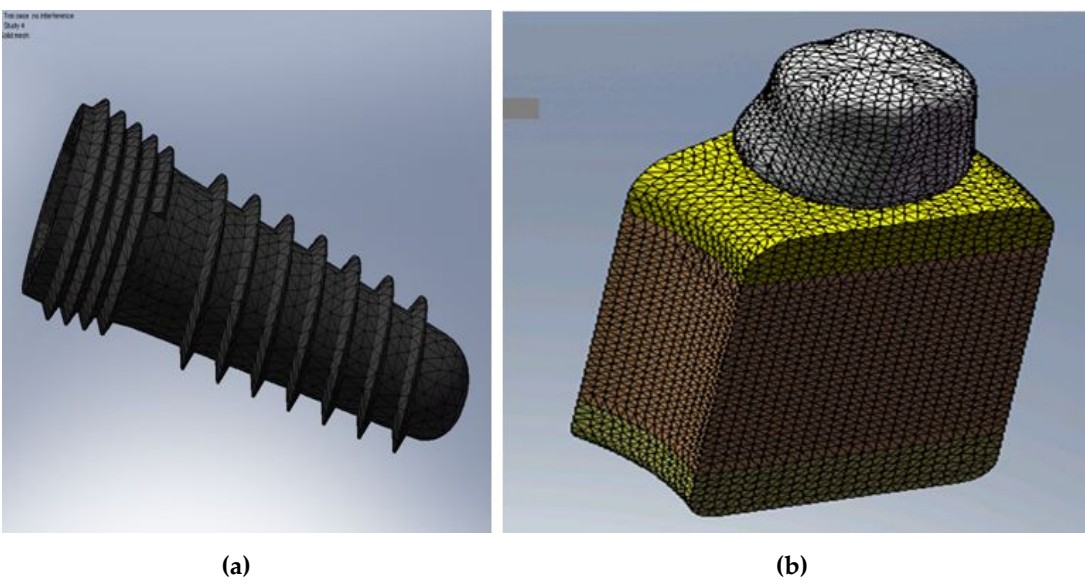

| (**a**) | (**b**) |

**Figure 4.** (**a**) FE model of the implant; and (**b**) FE model of the whole system.

In the FE model, there is a big number of finite elements due to the fact that there are fine structural elements, the mesh fineness appears as a necessity for a more realistic modeling of high-finesse constructive forms, such as threads and undercuts.

### 2.3. Contact Modeling

Osseointegration, as considered by a clinical point of view, refers more to the stability of the implant subjected to chewing loading and in close contact with the bone [22] rather than to the actual microscopic connection of bone tissue and implant surface. This connection is due to biological events which lead to the interaction of bone cells with implant surface after the surgical procedure.

The contact between the implant threads and bone is made on the thread sides. In our calculations, we considered a surface-to-surface contact (no penetration and preventing the interference between the implant and bone but allowing them to move away from each other to possibly form clearances) at the implant–one interface at the first stages of the osseointegration (lower values of E of the cortical bone), while a bonded contact, so without slipping, without friction at the final stage (the maximum E of the cortical bone). The FEM modeling was performed in order to capture the interaction between all the components of the implant and the bone, and the whole system was studied.

### 2.4. Material Models

In this section, the types of materials are presented, as are the main material constants for each component, i.e., bone (trabecular and cortical bone), implant and crown. These data are available from the literature [23] and from the data provided by the technical presentation of the implants used in this analysis. They are used as input data in the numerical calculations carried out in this study.

The most significant material constants used, respectively, for the implant, the abutment, the crown, trabecular bone and cortical bone are presented in Tables 1 and 2 [4] as follows:

**Table 1.** Material constants of the implant, its components and crown [4].

|  | Magnesium Alloy (Intermediate Part) | | Titan Alloy (Implant and Screw) | | Ceramic (Crown) | |
|---|---|---|---|---|---|---|
| Constant name | Value | Unit | Value | Unit | Value | Unit |
| Elastic modulus | $4.2 \times 10^{10}$ | N/m$^2$ | $1.048 \times 10^{11}$ | N/m$^2$ | $2.2059 \times 10^{11}$ | N/m$^2$ |
| Poisson Coefficient | 0.33 | | 0.31 | | 0.22 | |

**Table 2.** Material constants for the two types of bone [4].

|  | Trabecular Bone | | Cortical Bone | |
|---|---|---|---|---|
| Constant name | Value | Unit | Value | Unit |
| Elastic modulus | $1.8 \times 10^8$ | N/m$^2$ | $0.2 \times 10^9$–$1.8 \times 10^{10}$ | N/m$^2$ |
| Poisson coefficient | 0.3 | | 0.25 | |

### 2.5. Boundary Conditions

In general, in structural analysis, boundary conditions are put in displacements and/or forces in those regions of the structure where these entities are known. Zero displacement restrictions should be placed on some model boundaries to ensure equilibrium of the solution. In addition, restrictions must be set in nodes that are distant from the region of interest, which in our case, is the area surrounding the implant. It proceeds in this way to prevent the overlapping stress or strain field associated with the reaction forces, with the bone–implant interface.

In the FEM models considered in this paper, the lateral faces and bottom of the bone tissue are considered without displacement (nodes movements are fixed on those faces in all directions), (Figure 5a), considering that the influence of the loading no longer exists at that certain distance.

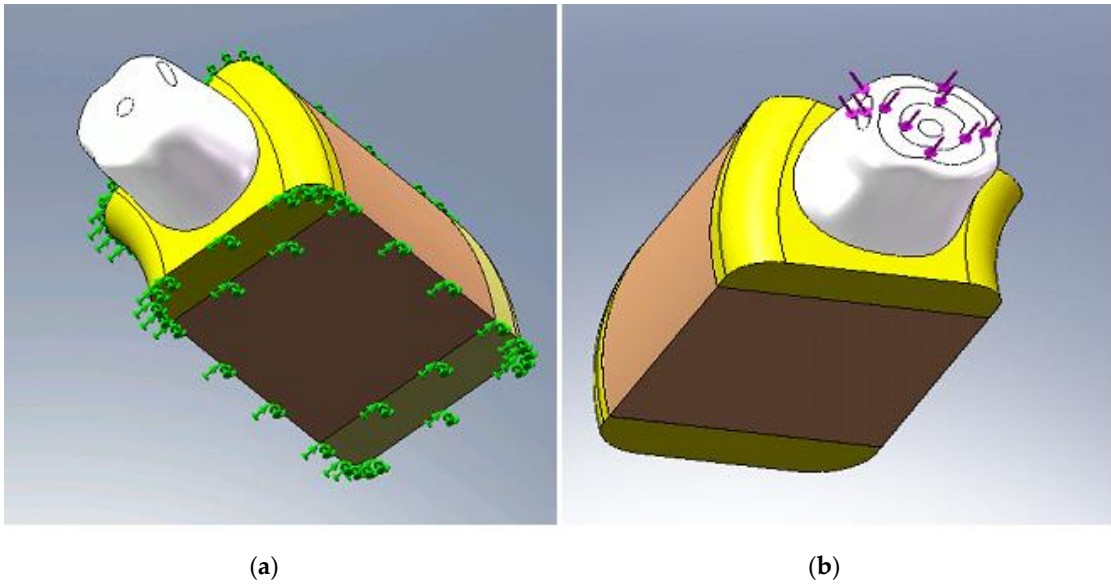

<div align="center">(<b>a</b>)　　　　　　　　　　　　　　　　　　　　　　　　　　　(<b>b</b>)</div>

**Figure 5.** (**a**) The boundary condition; and (**b**) applying axial and non-axial forces.

### 2.6. Applying Loads

Since the purpose of this study was to analyze osseointegration in progress, we considered applying a load of very low intensity to simulate the immediate loading of an implant of a jaw premolar, with a temporary crown for aesthetic reasons, in a lighter contact.

In the actual mastication, the repetitive pattern of cyclical forces [24] transmits the loading to the peri-implant bone via the dental implant. This determines the stress around the ridge and prosthetic structure.

Since the cyclic character of mastication is difficult to simulate [25], most FE studies use axial and/or non-axial forces, and a more appropriate simulation typically uses a combination of vertical and oblique forces (axial and non-axial). Non-axial loads generate a destructive stress especially in the cortical bone ridge in peri-implant bone region and clinically disturbing for prosthesis [26].

The masticatory force size can be variable depending on age, sex, edentulous, habits and may vary from anterior to posterior in the same mouth. The loads that simulate mastication forces generate stress concentration to be assessed and therefore should be considered an appropriate risk [27].

In this study, we used a realistic simulation of mastication forces by the simultaneous application of axial and non-axial loading with a range between 20 N–30 N on sufficiently large surfaces (Figure 5b).

Before analyzing the osseointegtation progress, convergence tests were performed, keeping the mesh refined until a little change in our solution was obtained.

### 3. Results

The FEM analysis usually calculates von Mises stress, i.e., equivalent stress, a scalar quantity that characterizes the amount of stress and that is very important in formulating criteria of damage, plasticity, fracture, etc. In this study, this was used to evaluate the effect of loading in the dental implant, prosthesis and surrounding bone.

The following will present the results of numerical calculation by FEM by figures representing von Mises stress, displacement and safety factor. In the figures below, the most favorable areas are the minimum values of stress or displacement, while areas with the greatest damage and high risk are characterized by higher values.

The safety factor is the ratio of admissible limit values and actual FEM calculated values of stress. The admissible limit values of stress are specific to each material and adopted in specific conditions

described in the literature. The figures representing the safety factor may indicate the necessity of appropriate design solutions adoption that eliminates risky zones that have a low safety factor.

In this study, we performed the calculations for different values of cortical bone Young's modulus, E. The corresponding values of Young's modulus, E lower ($E = 0.2 \times 10^8$ MPa) refers to the immediate phase after the implant insertion, which simulates a weaker biomechanical implantation medium, more damaged.

Increasing values for E means that a gradual osseointegration occurs. Achieving a value of E of intact bone is considered to signify the osseointegration process's completion.

We conducted two sets of calculations and summarized the results in the following two tables: Table 3 corresponds to set no.1, when the axial force is 20 N and the oblique force is 20 N, whereas Table 4 corresponds to the set no.2, when the axial force is 30 N and the oblique force is 30 N.

**Table 3.** Set no.1, the axial force is 20 N, and the oblique force is 20 N.

| Elasticity Modulus E | $\sigma_{\mathbf{max}}[\frac{N}{mm^2}]$ | $u_{res}[mm]$ | Minimum FOS |
|---|---|---|---|
| $1.8 \times 10^{10}$ | $7.987 \times 10^7$ (in cortical bone) | $4.309 \times 10^{-6}$ | 5.87 (in cortical bone) |
| $1.4 \times 10^9$ | $7.798 \times 10^7$ (in cortical bone) | $5.25 \times 10^{-6}$ | 6.006 (in crown) |
| $1.0 \times 10^9$ | $6.897 \times 10^7$ (in implant) | $5.77 \times 10^{-6}$ | 6.893 (in crown) |
| $0.6 \times 10^9$ | $8.044 \times 10^7$ (in implant) | $7.17 \times 10^{-6}$ | 6.893 (in crown) |
| $0.2 \times 10^9$ | $12.66 \times 10^7$ (in implant) | $10.01 \times 10^{-6}$ | 6.893 (in crown) |

**Table 4.** Set no.2, the axial force is 30 N, and the oblique force is 30 N.

| Elasticity Modulus E | $\sigma_{\mathbf{max}}[\frac{N}{mm^2}]$ | $u_{res}[mm]$ | Minimum FOS |
|---|---|---|---|
| $1.8 \times 10^{10}$ | $1.288 \times 10^8$ (in cortical bone) | $6.829 \times 10^{-6}$ | 3.668 (in cortical bone) |
| $1.4 \times 10^9$ | $1.173 \times 10^8$ (in cortical bone) | $7.853 \times 10^{-6}$ | 4.015 (in cortical bone) |
| $1.0 \times 10^9$ | $1.033 \times 10^8$ (in implant) | $9.832 \times 10^{-6}$ | 4.595 (in crown) |
| $0.6 \times 10^9$ | $1.288 \times 10^8$ (in implant) | $11.93 \times 10^{-6}$ | 4.595 (in crown) |
| $0.2 \times 10^9$ | $1.899 \times 10^8$ (in implant) | $17.51 \times 10^{-6}$ | 4.595 (in crown) |

Tables 3 and 4 present the values of the main quantities calculated during the various stages of osseointegration as simulated in our numerical model, namely considering the damaged properties of cortical bone immediately after implant surgery, and further, as osseointegration progresses, cortical bone becomes stronger, so that in the end its mechanical properties are identical to those of intact bone. The total displacement or resulting displacement $u_{res}$ is calculated by the formula:

$$u_{res} = \sqrt{u_x^2 + u_y^2 + u_z^2}$$

where $u_x$, $u_y$, $u_z$ are the displacements along the axes $x,y,z$, while the factor of safety (FOS) is calculated by the ratio allowable stress/calculated stress. The calculated stress can be any considered stress: von Mises (the most used), Tresca and others. As a rule, the calculated stress is that corresponding to a failure criterion. We used von Mises and Tresca criterion and the differences were negligible. We present in this paper the values obtained by the von Mises criterion.

The following figures present the cases of maximum values of E ($8 \times 10^{10}$ Pa), corresponding to a full osseointegration and the case of minimum values of E ($0.2 \times 10^{10}$ Pa) corresponding to a time immediately after implant insertion, and therefore an early osseointegration as well.

Thus, Figure 6a,b represent the von Mises stress distribution for E minimum (a) and maximum (b), respectively.

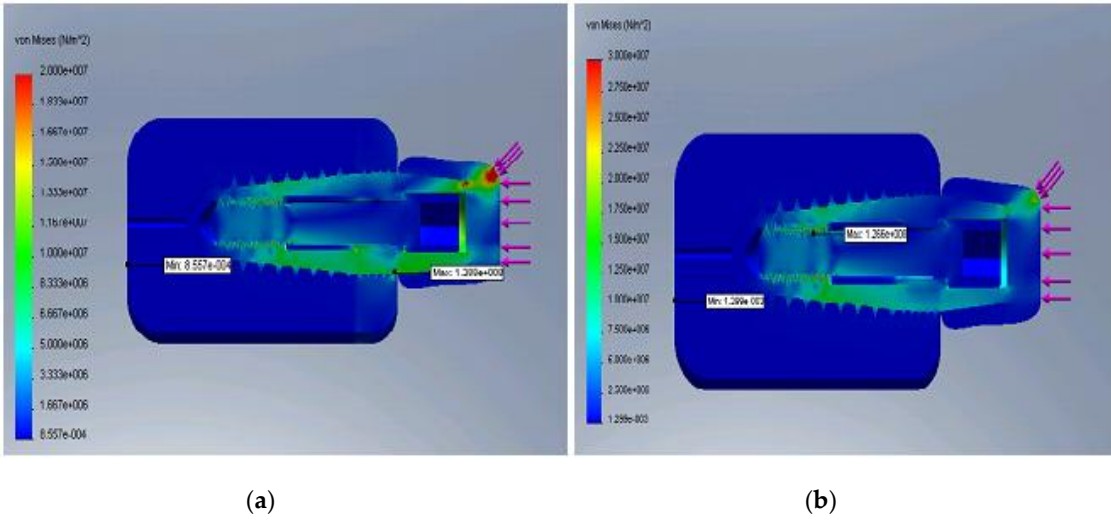

(**a**)　　　　　　　　　　　　　　　　　　　　　　　(**b**)

**Figure 6.** Distribution of the von Mises stress for the minimum E (**a**) and maximum E (**b**).

Figure 7a,b represent the displacement distribution for minimum E (a) and maximum E (b), respectively.

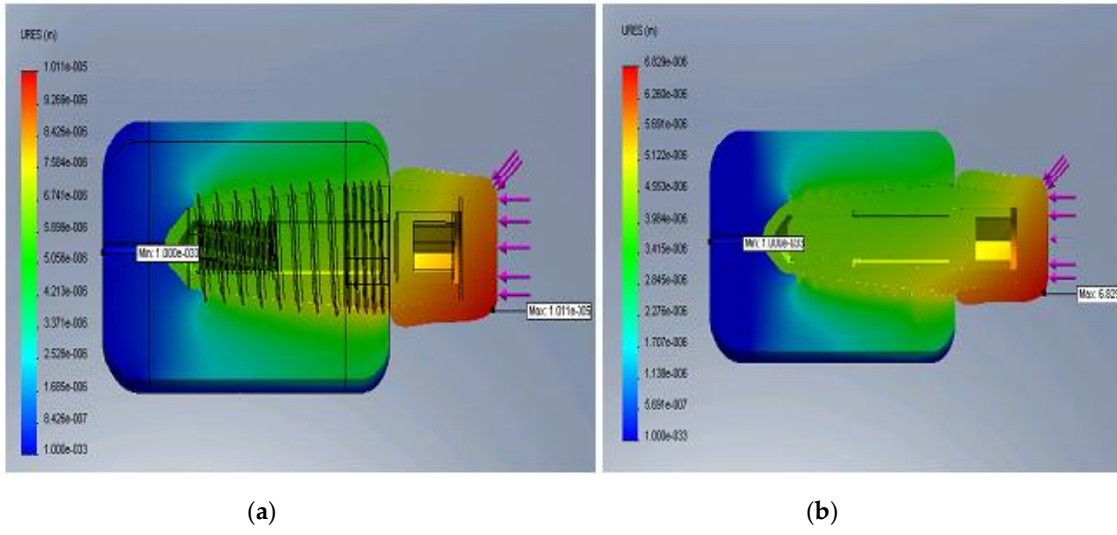

(**a**)　　　　　　　　　　　　　　　　　　　　　　　(**b**)

**Figure 7.** Displacement distribution for the minimum E (**a**) and maximum E (**b**).

Figure 8a,b represent the safety factor distribution for E minimum (a) and maximum (b), respectively.

It was observed that the rigid implant distributed the bending stress to its peak, and as the structure became stiffer, the cortical bone began to take over the bending loading. Displacements decreased as the bone–implant structure was reinforced and the most critical zone was within the upper cortical bone.

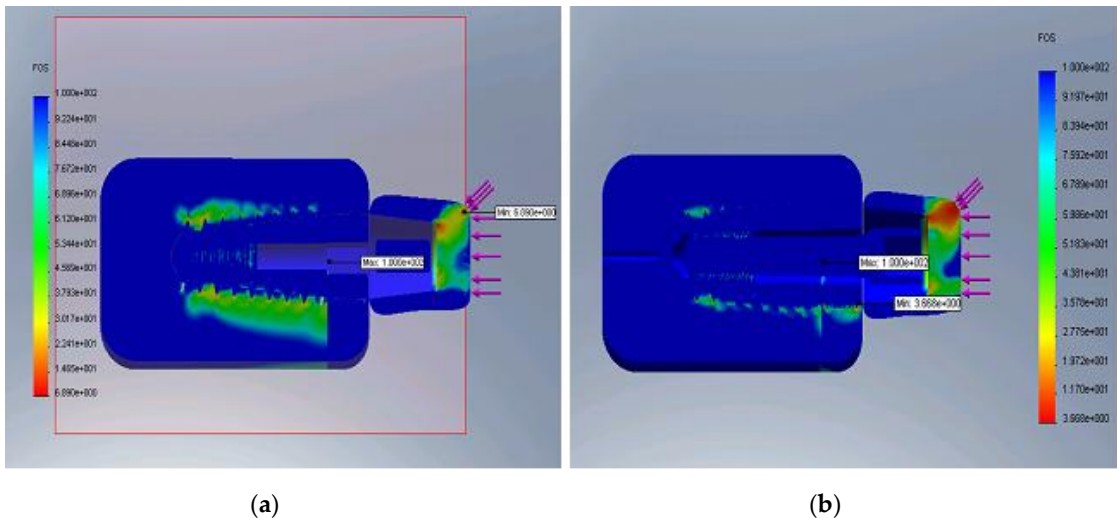

(**a**)                    (**b**)

**Figure 8.** Distribution of the safety factor for the minimum E (**a**) and maximum E (**b**).

## 4. Discussion

The results obtained in this study were in full compliance with the clinical practice and with the results from the literature [5,9,10,28], as it follows. As implant osseointegration develops, the rigidity of the peri-implant bone grows, so that in the end (healing) we can observe that the cortical bone is the one that provides the rigidity and stability of the implant, and the cortical bone stress reaches highest values, placed at the level of the mandibular ridge. During the period after the osseointegration process, the displacements are lower and they are placed only in the cortical bone area [1,29].

The oral implant insertion implies the incurrence of a microcraking field in the bone tissue at the implant–bone interface [30]. This leads to the mechanical properties' damage of the cortical bone and the decrease in the bone tissue stiffness around the implant. This is illustrated by the stress field aspect which is characterized by low stress values and their distribution extends along the implant [9]. It also finds that the bone–implant interface displacement has the highest values [10].

The limitations of the study were caused by several aspects, as it follows.

Osseointegration is a growth-increased change of bone structure between implant surface and bone tissue, and in the change of the osseointegration were included both the structure distribution and bone quantity to reflect the stability of the dental implant. This study partially took into account the complexity of the osseointegration progress modeling.

The results of the FEM analysis cannot be implemented directly in clinical situations, but one can design a model such that to simulate a real situation as well as possible [4]. Simplifications and method assumptions are, however, some limitations of studies using FEM.

FEM analysis should be interpreted carefully. In most cases, the numerical studies of oral implantology, e.g., using homogeneous, elastic and isotropic materials, which remain after the loading to which they are subjected, do not reflect the real situation. Taking into account these issues, it would require very laborious calculations and complicated laboratory tests to determine the material constants for a biological material, which behaves as nonlinear, heterogeneous, etc. [31].

As the cortical layer is an important structure that affects the stress distribution and stress shielding for a dental implant, the thickness of the cortical layer should be considered as a study issue, for instance. The effect of the cortical layer would be discussed as a relationship between the osseointegration progress and the biomechanical influence. On the other hand, to work with a simpler model, we adopted damaged mechanical properties for the entire cortical bone layer in the geometric model for the initial period after implant surgery, although for a more realistic approach, only a smaller area around the implant.

We used the von Mises stress as a stress representation, considering it rather a convenient one, rather than an appropriate one, especially for bone and ceramic materials that are not at all ductile as the von Mises criterion requires. Moreover, as the trabecular bone possesses weaker mechanical properties, it plays a secondary role in approaching the progress of osseointegration from a biomechanical point of view, so that the effect on the calculations would be negligible.

The problem domain could be considered small, but enough to have a correctly defined problem. The domain is chosen correctly, as long as the interaction of the neighborhood does not exist. That was proven in our model by the results and can be observed in the figures representing the stress, for instance. The fact that we obtained almost zero stress at boundaries shows the correctness of the choice of the domain, as well as of the boundary conditions. The influence of implant surgery is small and does not affect large areas of bone.

We also mentioned as possible limitations of the study the shape of the dental crown and the fact that it is directly in touch with the cortical bone that could lead physiologically to the formation of the well known biological space of at least 2–3 mm. We mention, however, that it was considered a provisional crown for a period of several months, so that the impact can be negligible.

The contact issue in the models focusing on the osseointegration progress is a delicate one. We used in our calculations a surface-to-surface contact at the interface implant–bone during the first stages of the osseointegration, and a bond contact at the final stage of it, for a maximum E of cortical bone.

On the other hand, the FEM model represents a static situation for a certain time and loading, and does not represent an actual clinical situation. Moreover, the system loading is rather dynamic and cyclical.

In addition, osseointegration is a complex process, implying both biological and mechanical factors [24], so that the FEM represents only a certain segment of the analyze.

Therefore, the results by FEM must be correlated with preclinical and clinical long-standing studies in order to obtain their validation.

The numerical model under consideration must run for as many scenarios, e.g., for different values of loads, their various plausible locations, for various values of biological material.

The numerical model requires a thorough validation, which is mandatory and it is done [32] either by the comparison of results for the same scenario using several numerical codes or standard FE programs; or comparing the results, at least qualitatively, with real clinical situations and the comparison of numerical results with the results obtained by available laboratory tests. In this study, we conducted a thorough validation of the criteria outlined previously.

Note that we made several versions of the mesh to a fineness which does not lead to errors greater than a 3.5% solution [33]. Element convergence tests were performed in order to obtain a reliable FE model to study the osseointegration progress.

## 5. Conclusions

In this article, an original approach using FEM calculation was used to model the osseointegration process progress. The stress and displacement in the dental implant and surrounding bone was calculated, being used to assess the biomechanical behavior of the whole assembly made up of implant system, ceramic crown and surrounding bone.

The novelty of this study included, firstly, the achievement of an accurate modeling of the bone–implant–crown system, particularly in terms of the implant (the geometrical model is an exact replica of a real implant) and secondly, the simulation in a realistic way of the osseointegration progress by taking into account the gradual increase in the damaged mechanical properties of the cortical bone. The model also includes a realistic simulation of loadings, simultaneously containing vertical and oblique forces acting on a certain surface of the crown.

The results obtained by FEM are complementary to other clinician information and they should be related to preclinical and long-term clinical trials for conclusions and right decisions.

**Author Contributions:** Conceptualization, I.R. and S.R.; methodology, I.R.; software, S.R.; validation, I.R. and S.R.; formal analysis, S.R.; investigation, I.R.; resources, I.R.; data curation, I.R.; writing—original draft preparation, S.R.; writing—review and editing, I.R. and S.R.; visualization, S.R.; supervision, I.R.; project administration, I.R. All authors have read and agreed to the published version of the manuscript.

**Funding:** This research received no external funding.

**Acknowledgments:** We wish to thank E. Avram E. for unconditional support for the realization of the geometrical model and the FEM calculation. Both thank to V. Năstăsescu for high professionalism and expertise on FEM use.

**Conflicts of Interest:** The authors declare no conflict of interest.

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
