# Peer review of "Modeling of Dental Implant Osseointegration Progress by Three-Dimensional Finite Element Method"

_applsci, doi:10.3390/app10165561_

Round 1

Reviewer 1 Report

This paper developed a FEM model of an dental implant in bone as it osseointegrates. They estimated the range in elastic modulus of bone from 0.1 e8 Pa after osteotomy and implant placement to 1.8 e8 Pa after osseointegration has stabilized.

They did a nice job of matching the implant geometry and crown to clinical geometries, but their reported data was not reported clearly and hard to interpret. The values used for the model lacked sufficient justification, and there was no mention of how those parameters correspond to time after implant placement. For example, 1 month vs 6 months? How are these values clinically relevant.

  1. does this model correspond to a immediate loading of an implant? I am confused by this statement, "Since the purpose of the study is to analyze osseointegration in progress, we considered applying load of very low intensity to simulate the osseointegration period, only after full implant osseointegration it would facilitate the use of immediate and early loading protocols.” Does this model take into account immediate loading? Including the crown suggests that the implant will be loaded and in occlusion. How does this apply “very low intensity of loading” if the implant is in occlusion?
  2. What is the justification for an elastic modulus of 0.2 x10^8 Pa immediately after implant insertion?
  3. Units on Lines 227-228 are inconsistent. They should be Pa instead of MPa, since they are reported as x10^8.
  4. The colors of the figures are hard to interpret, because a different scaling is used for parts A and B. This is misleading and hard to compare between early and maximum E. The color grading should be consistent within both panels. Also the text on these figures are extremely hard to read and should be added in a vectorized format so the text is legible.
  5. Authors should discuss why the displacement values are higher in the ceramic crown than the implant-bone interface. I find this implausible, especially at the early minimum E. This result indicates to me that the initial E might be too stiff relative to the native properties after implant placement. Further, their model is static and ignores creep properties of bone, which could cause deformation over time.

Author Response

This paper developed a FEM model of an dental implant in bone as it osseointegrates. They estimated the range in elastic modulus of bone from 0.1 e8 Pa after osteotomy and implant placement to 1.8 e8 Pa after osseointegration has stabilized.

They did a nice job of matching the implant geometry and crown to clinical geometries, but their reported data was not reported clearly and hard to interpret. The values used for the model lacked sufficient justification, and there was no mention of how those parameters correspond to time after implant placement. For example, 1 month vs 6 months? How are these values clinically relevant.

  1. does this model correspond to a immediate loading of an implant? I am confused by this statement, "Since the purpose of the study is to analyze osseointegration in progress, we considered applying load of very low intensity to simulate the osseointegration period, only after full implant osseointegration it would facilitate the use of immediate and early loading protocols.” Does this model take into account immediate loading? Including the crown suggests that the implant will be loaded and in occlusion. How does this apply “very low intensity of loading” if the implant is in occlusion?

Yes, it's an immediate loading of an implant in the first 6 months.

The model under consideration in our study was that of a premolar on the jaw (from a regrettable error it was mentioned mandible), with a temporary crown for aesthetic reasons, in a lighter contact.

The mentioned phrase was remedied and in addition we specified the type of tooth for which the implant surgery took place.

  1. What is the justification for an elastic modulus of 0.2 x10^8 Pa immediately after implant insertion?

Justification is that immediately after implantation, the bone is considered damaged, dysfunctional, then, by starting the osseoformation, it enters a process of gradual adaptation, of preparation for taking over the forces of mastication. The value was thought to be less than the native one for the cortical bone to simulate the bone damage caused by the implant surgery.

  1. Units on Lines 227-228 are inconsistent. They should be Pa instead of MPa, since they are reported as x10^8.

Indeed. Done.

  1. The colors of the figures are hard to interpret, because a different scaling is used for parts A and B. This is misleading and hard to compare between early and maximum E. The color grading should be consistent within both panels. Also the text on these figures are extremely hard to read and should be added in a vectorized format so the text is legible.

Done.

  1. Authors should discuss why the displacement values are higher in the ceramic crown than the implant-bone interface. I find this implausible, especially at the early minimum E. This result indicates to me that the initial E might be too stiff relative to the native properties after implant placement.

One can simply think to the fact that the tooth top displacement is bigger than root tooth displacement subjected to a certain load or movement.

From a mechanically point of view, this model can be assimilated to the model of a clamped beam subjected to compression and bending.

Further, their model is static and ignores creep properties of bone, which could cause deformation over time.

The complexity of the osseointegration process means that the modeling cannot cover the whole multitude of aspects involved.

Modeling the behavior of the implant taking into account the creep and relaxation properties may be the purpose of another paper. Any numerical analysis of some structures, especially complex ones (by geometry and materials), is recommended to be studied first of all in static regime. If it provides necessary and useful data for the proposed purpose, static analysis may be sufficient. Agree that no research is completely completed; often new aspects appear to be researched, but the researcher decides for a work if his numerical investigation continues or if the one carried out meets the declared purpose of the work.

The model presented in the article represents an approximation of the process and which, for the moment, does not take into account all the aspects related to osseointegration, but which can be, undoubtedly, the object of future calculations.

Reviewer 2 Report

The aim of the study could be interesting if the only aim is considered to provide a FEM analysis who can describe the stress on the bone during the osseointegration which can be partially simulated, by an increase of elasticity modulus. However, have to be well underlined that as described in the discussion section as a limit, these data can't be easily connected with the clinical reality because many biological, not well-known factors influence the osteointegration.

The main limit of the study is the model that is defined as realistic but it isn’t.

First, the dental crown is directly in touch with the cortical bone. This is an unrealistic situation that leads physiologically to the formation of the well known biological space of at least 2-3mm. This space could affect FEM measurements.

The crown has an unreal shape. It is shaped as a truncated cone when a standard dental crown is more prominent and the cuspids create cantilever. This very often creates an overloading problem. For this reason, have to be considered in the FEM analysis

The prosthetic materials thickness is not declared and appears from the figure, not realistic. This parameter can also affect the results.

Finally, the manuscript has to be revised. Especially the initial parts of the M&M and Results sections are transferred to the introduction or discussion or erased if redundant.

Author Response

The aim of the study could be interesting if the only aim is considered to provide a FEM analysis who can describe the stress on the bone during the osseointegration which can be partially simulated, by an increase of elasticity modulus. However, have to be well underlined that as described in the discussion section as a limit, these data can't be easily connected with the clinical reality because many biological, not well-known factors influence the osteointegration.

Indeed. The complexity of the osseointegration process means that the modeling cannot cover the whole multitude of aspects involved. The model presented in the article represents an approximation of the process and which, for the moment, does not take into account all the aspects related to osseointegration, but which can be, without a doubt, the object of future calculations.

Moreover, indeed, in general, FEM studies dedicated to dentistry cannot be directly validated by clinical cases, but only by laboratory experiments if the models are suitable for it.

It remains also, their validation through calculations made with other programs for the same boundary problem. But obviously, in advance extensive convergence studies must be achieved, that we carefully performed.

The main limit of the study is the model that is defined as realistic but it isn’t.

If we agree with the complexity of the subject, if we agree with the existence of many factors about which we do not know or know very little, then we should agree that a numerical model that captures a visible phenomenon is welcome, even if the determined quantities may be relative in value, but simulate the phenomenon.

The numerical model used is close to reality and very useful, although it could be improved by increasing the wealth of concepts and notions of mechanical engineering, contact mechanics, damage, failure, technology and obviously numerical modeling.

In our paper, for instance, we succeeded in creating a geometric model that was very close to reality, in dimensions, shapes, etc… On the other hand, and in terms of loading, we simulated the masticatory forces as well as possible (axial and oblique forces acting on a relatively big crown area).

First, the dental crown is directly in touch with the cortical bone. This is an unrealistic situation that leads physiologically to the formation of the well known biological space of at least 2-3mm. This space could affect FEM measurements.

Indeed.

It is true that in the model made the distance between the dental crown and the cortical bone is not 2… 3 mm, but it should be noted that there is no contact between them. The figures, both for the stresses and displacement, highlight this. If there was contact, locally, in the contact points, other colors of the stress field appeared. Moreover, our aim is to study the osseointegration progress which could be limited to several months and the mentioned situation effect might be not so dominant. But in future models, definitely, that space would be taken into account. Moreover, in clinical cases, the crown is provisional during the osseointegration period, and only for aesthetic reasons.

The crown has an unreal shape. It is shaped as a truncated cone when a standard dental crown is more prominent and the cuspids create cantilever. This very often creates an overloading problem. For this reason, have to be considered in the FEM analysis

Very true. As our study considers immediate loading, which is quite low, 20-30 N and a provisional crown, that shape could be tolerated.

The prosthetic materials thickness is not declared and appears from the figure, not realistic. This parameter can also affect the results.

The crown thickness varies between 0.7-1.5 mm. This a temporary crown and we focused on the osteointegration issues.

Finally, the manuscript has to be revised. Especially the initial parts of the M&M and Results sections are transferred to the introduction or discussion or erased if redundant.

Done.

Reviewer 3 Report

The manuscript presents a very detailed CAD model of a dental implant and a FEM analysis of load transmission during mastication, taking into account trabecular bone damage before osseointegration.

The model of the mandible however is very crude, and the simplifying assumption adopted should be discussed and their effect on the results assessed.

Major concerns

1. The first and most significant problem is the very small section around the dental implant considered, and the assumption that at such a small distance "the state of stress and strain is not influenced" by the dental implant (line 96). Moreover imposing a fixed boundary condition on this artificial boundary (line 160) is equivalent to the assumption of a very stiff nearby bone, which is not physical.

It is very plausible that the stress/strain state on the dental implant is affected by the presence of the roots of nearby teeth: in other words mechanical interactions between nearby teeth should not be ruled out a priori.  The authors should detail the reason for which it is not necessary to model a larger section of the mandible and the nearby teeth, and asses which is the effect of the fixed boundary conditions imposed on the lateral boundary. E.g a sensitivity analysis with respect to the dimensions of the section considered would be very useful. 

2. Trabecular bone is a very complex material: both form the point of view of its elastic properties and its strength. Bone is in fact anisotropic, and using the von Mises stress criterion for assessing its strength (line 201--204) is grossly wrong. (Von Mises stress was introduced for defining the yield locus of a ductile metal; although sometimes it is sensible to use it for different materials, this is not true for trabecular bone.) See e.g. BIOMECHANICS OF TRABECULAR BONE by Keaveny et. al. Annu. Rev. Biomed. Eng. 2001. 3:307–33.

The authors should discuss the reasons of adopting an isotropic material, and why the damage is constant in the whole section of the modelled mandible (brown section in fig. 3.(b)). I would assume that the damage is greater near the implant, but less important on the artificial boundary, which should be far away from the implant itself. By the way, this artificial boundary is modelled as perfectly rigid, i.e. by imposing fixed boundary conditions, and this seems to contradict the assumption that the bone is damaged up to this boundary.

Moreover von Mises stresses should be presented only for the metallic parts of the implant. For the trabecular bone, results in terms of principal stresses (max and min) and also principal strains (max and min) should be presented and discussed.

In fact the bone has a completely different strength in traction with respect to compression, and the von Mises equivalent stress (which is by definition always positive) can not discriminate between traction and compression. On the contrary principal stresses and strains are signed quantities, and therefore allow the identification of zones of predominant compression or traction. 

Damage and fracture of bone is a very complex phenomenon, so it makes no sense to present a FOS (factor of safety) plot (fig. 8 of the manuscript.) without defining with respect to which damage or fracture criterion it is defined. Unless complex criteria are adopted (e.g. Tsai-Wu.) it is better to omit a FOS estimation for the bone.

3. The conclusions, when point 1. and 2. are clarified, should go into a deeper analysis of the FEM results, possibly not only from a qualitative point of view, but also by some more quantitative considerations. In fact the final claim "The results obtained by FEM are complementary to other clinician information and they should be related to preclinical and long-term clinical trials for conclusions and right decisions" (line 310--311) appears to me not substantiated: how should the clinician use the FEM results to asses a dental implant? Stresses and displacement presented in tables 5 and 6 are acceptable or not? Which information do we draw from these figures?

Minor concerns.

4. From the point of view of a FEM expert the description of the contact model adopted (lines 122–134) is insufficient. Please give more details on the modelling assumptions (unilateral contact? separation allowed? hard contact or a "softened" one? is sliding modelled? friction o frictionless?) and contact algorithm (eg. penalty, ...)

Please elaborate the sentence "It is considered that all threaded areas have intrinsic backlashes" (line 134) which is a little obscure to me. Does this means that there is a clearance between the screw threads? Can this clearance please be quantified? I would expect that any backlash in the thread would be negligible, once tightened. It therefore not clear what you are modelling.

Moreover why is the stress due tightening the threaded components negligible?

5. Lines 56–59: please state explicitly the version of the adopted programs and on which computing platform the computations were performed (pc or workstation, OS version, RAM size, etc.)

6. Line 101, fig. 3: please clarify the color coding of the bone tissue. It may not be obvious for a non medical reader which one (yellow or brown) is trabecular or cortical tissue.

Author Response

The manuscript presents a very detailed CAD model of a dental implant and a FEM analysis of load transmission during mastication, taking into account trabecular bone damage before osseointegration.

In our FEM calculations we took into account the cortical bone damage as we stated in the paper (e.g in abstract: This process is simulated by taking into account gradual increasing of the damaged biomechanical properties of the cortical bone. The results reveal that as the implant osseointegration occurs gradually, the bone stiffness from the peri implant area increases gradually, such that in the end (healing) we observe that the cortical bone begins to take over the bending loading. In addition, the displacements decrease as the osseointegration gradually occurs and the cortical bone stress reaches higher values, ​​which are placed in the mandibular ridge.).

BUT, you are right, even if in table 2 we set E=0.2´109-1.8´1010, in tables 5 and 6, modified now in 3 and 4, it appears from a typing error E=0.2-1.8´108, which coud be order of magnitude of values of trabecular bone! We are very sorry about that!

Our FEM calculations are performed for cortical bone values, i.e. E=0.2´109-1.8´1010! That is our estimation for cortical bone damage caused by the implant surgery.

The model of the mandible however is very crude, and the simplifying assumption adopted should be discussed and their effect on the results assessed.

Our model is a jaw portion, as our model refers to jaw premolar implant and a provisional crown, for aesthetic reasons, subjected to an immediate loading and in a lighter contact. Our focus has been turned to osseointegration progress occurring in the first months from implant insertion.

From a regrettable mistake we mentioned mandible automatically everywhere in the paper.

Major concerns

  1. The first and most significant problem is the very small section around the dental implant considered, and the assumption that at such a small distance "the state of stress and strain is not influenced" by the dental implant (line 96). Moreover imposing a fixed boundary condition on this artificial boundary (line 160) is equivalent to the assumption of a very stiff nearby bone, which is not physical.

In principle, the observation is correct, but in this case it is not applicable. This is proved by the field of stress and displacement, which show, in all cases, extremely low values, virtually zero and certainly negligible, compared to those in the vicinity of the implant. Those images show that the model could consider even a smaller area in the vicinity of the implant.

You are right, the domain is too small. The restrictions must be placed in the nodes that are at a distance from the region of interest, in our case, the area around the implant. This is done to prevent overlap of the stress field associated with the reaction forces with the bone-implant interface. Usually, for a single element FEM study, the most works consider such a domain.

Considering a larger domain with the larger neighborhood, containing also the vicinity teeths would represent a big volume of extra data, but it could be considered in future works. Our calculation is correct as we obtained a stress field which tends to zero at mentioned boundaries.

It is very plausible that the stress/strain state on the dental implant is affected by the presence of the roots of nearby teeth: in other words mechanical interactions between nearby teeth should not be ruled out a priori.  The authors should detail the reason for which it is not necessary to model a larger section of the mandible and the nearby teeth, and asses which is the effect of the fixed boundary conditions imposed on the lateral boundary. E.g a sensitivity analysis with respect to the dimensions of the section considered would be very useful. 

Zero displacement restrictions must be placed on some boundaries of the model to ensure solution balance. In support of this idea is the fact that for example, in Figures 6a and b representing the von Mises stress, we observe the almost zero stress values at the edges of the domain, which indicates the balance and that the studied process does not influence and is not influenced by neighborhoods.

  1. Trabecular bone is a very complex material: both form the point of view of its elastic properties and its strength. Bone is in fact anisotropic, and using the von Mises stress criterion for assessing its strength (line 201--204) is grossly wrong. (Von Mises stress was introduced for defining the yield locus of a ductile metal; although sometimes it is sensible to use it for different materials, this is not true for trabecular bone.) See e.g. BIOMECHANICS OF TRABECULAR BONE by Keaveny et. al. Annu. Rev. Biomed. Eng. 2001. 3:307–33.

The anisotropic character of the bone was not denied, but for the study, the homogeneous and isotropic bone hypothesis was adopted, as a working covering hypothesis and at the same time easier to model. We assure you that in other studies in which the objective would be the analysis of the bone, the anisotropic bone model would have been used. The discussion is complicated, if we consider the specific diversity of the person under treatment, the exact knowledge of the directions of anisotropy and the properties on those directions. Then what kind of anisotropy? General, with what peculiarities, orthotropy, what is the orientation of the directions, etc.? These issues cannot be taken into account exactly and for this reason the adoption of simplified assumptions or models is welcome, especially if they correspond to the proposed purpose and are comprehensive, without answering to some actual issues but not significant to the intended purpose.

On other hand, in our calculations the cortical bone plays an important role, considering that its initial damage and then gradually increasing its strength may simulate in a certain way the osseointegration progress.

The authors should discuss the reasons of adopting an isotropic material, and why the damage is constant in the whole section of the modelled mandible (brown section in fig. 3.(b)). I would assume that the damage is greater near the implant, but less important on the artificial boundary, which should be far away from the implant itself. By the way, this artificial boundary is modelled as perfectly rigid, i.e. by imposing fixed boundary conditions, and this seems to contradict the assumption that the bone is damaged up to this boundary.

We totally agree that most biologic materials are anisotropic, but for certain reason of simplifying the model and focusing our calculations to other issues, it is quite common to set elastic, homogeneous and isotropic properties for the biologic materials.

Moreover, setting non-elastic, non-homogeneous or anisotropic properties involves highly accurate and complicated laboratory tests that must provide the material constants in these cases.

The adopted model considering the same value of E in the entire cortical bone layer is comprehensive in terms of general conclusions. This assumption reduced considerably the effort of modeling and calculation.

The perfectly rigid boundary modeling was also adopted for calculation reasons, given that the number of nodes exceeds one million. But this hypothesis is comprehensive and correct, because the field of stresses and displacements show that on the boundary and in its wide vicinity the values ​​are or tend to zero. So the model is correct.

Moreover von Mises stresses should be presented only for the metallic parts of the implant. For the trabecular bone, results in terms of principal stresses (max and min) and also principal strains (max and min) should be presented and discussed.

The use of von Mises theory can be questionable (even inappropriate) only in the case of uniform triaxial states of compression, for which the surface of the limit states is a closed surface.

The model used and the loading mode highlight the main requirement, that of compression and bending, which fully corresponds to reality.

In fact the bone has a completely different strength in traction with respect to compression, and the von Mises equivalent stress (which is by definition always positive) can not discriminate between traction and compression. On the contrary principal stresses and strains are signed quantities, and therefore allow the identification of zones of predominant compression or traction. 

Indeed. The Von Mises stress is in this context is a stress measure (always positive indeed), the second stress invariant

and it can be calculated for any material. If we want to study the stress pattern for the whole system bone-implant-crown we must set to an unique representation.

Damage and fracture of bone is a very complex phenomenon, so it makes no sense to present a FOS (factor of safety) plot (fig. 8 of the manuscript.) without defining with respect to which damage or fracture criterion it is defined. Unless complex criteria are adopted (e.g. Tsai-Wu.) it is better to omit a FOS estimation for the bone.

It is true. In fact we performed the calculation of FOS both with respect to Tresca and Von Mises criterions and we found really negligible differences. In the paper we presented FOS pattern with respect to Von Mises criterion.

Of course, the presentation of the FOS could be missing, but it was presented as a conclusion to validate the study under the given conditions, in an easy and expressive presentation.

  1. The conclusions, when point 1. and 2. are clarified, should go into a deeper analysis of the FEM results, possibly not only from a qualitative point of view, but also by some more quantitative considerations. In fact the final claim "The results obtained by FEM are complementary to other clinician information and they should be related to preclinical and long-term clinical trials for conclusions and right decisions" (line 310--311) appears to me not substantiated: how should the clinician use the FEM results to asses a dental implant? Stresses and displacement presented in tables 5 and 6 are acceptable or not? Which information do we draw from these figures?

Yes, you are talking here about the general limitation of a FEM study and its connection with clinical dentistry. We obtain by FEM analysis mainly qualitative responses that must be harmonized with clinical studies findings. Some FEM analyses simulate some laboratory tests and in these cases we can talk about quantitative matters.

The authors dialogue, between on one hand a scientist and clinician in the same time, and on the other hand a scientist with expertise in numerical calculations, together with the whole team of mechanical engineers, provided the harmony between calculations and clinical practice. This is not a direct link, but a

Yes, the results obtained were thoroughly analyzed and they are totally acceptable from a clinician point of view.

Moreover they are valuable: for the primary osseontegration period of about 0-6 months the clinicians get the information on the optimal period of loading to crown mounting, and for the second osseointegration period of about 6-12 months, in order to establish the optimal staging of the controls for the integration of the implant in terms of the functionality of the entire system of bone-implant-crown.

The use of the results of this study is a matter depending on the reader, on the clinical dentist. The work is not about verdicts, nor does it exhaust the discussions, but if they are provoked between specialists, the work has at least partially reached its goal.

Minor concerns.

  1. From the point of view of a FEM expert the description of the contact model adopted (lines 122–134) is insufficient. Please give more details on the modelling assumptions (unilateral contact? separation allowed? hard contact or a "softened" one? is sliding modelled? friction o frictionless?) and contact algorithm (eg. penalty, ...)

Please elaborate the sentence "It is considered that all threaded areas have intrinsic backlashes" (line 134) which is a little obscure to me. Does this means that there is a clearance between the screw threads? Can this clearance please be quantified? I would expect that any backlash in the thread would be negligible, once tightened. It therefore not clear what you are modelling.

The contact between the implant threads and bone is made on the thread sides. Due to the tightening resulting at the end of the screwing operation, the contact can be bonded, so without slipping, without friction. The model captures the end of the insertion procedure, so there is no clearance. The FEM modeling was performed in order to capture the interaction between all the components of the implant and the bone, the whole system was studied.

Moreover why is the stress due tightening the threaded components negligible?

The difference between the modulus of elasticity of the bone and the implant is large, so that at the end of implant surgery, the bone in the vicinity of the implant thread suffered plastic strains, fractures and microcracks, and the stress is much lower than that of the implant.

  1. Lines 56–59: please state explicitly the version of the adopted programs and on which computing platform the computations were performed (pc or workstation, OS version, RAM size, etc.)

The calculation was performed by the Cosmos program integrated in the SolidWorks product on PC, 16 M RAM and Intel i7 processor, Windows OS. We intentionally omitted a lot of technical details as they are not interesting for most readers. Some reviewers suggested even removing some other calculation details, keeping only the significant ones.

  1. Line 101, fig. 3: please clarify the color coding of the bone tissue. It may not be obvious for a non medical reader which one (yellow or brown) is trabecular or cortical tissue.

Done. Bone tissue with the layers of cortical bone (yellow) and trabecular bone (brown)

Round 2

Reviewer 1 Report

The authors have sufficiently addressed my comments. Thank you.

Author Response

Thank you for your comments and appreciation.

Reviewer 2 Report

I agree with the complexity of the subject but the model shown, as previously indicated, has some specific limitations which have to be clearly described.
I can understand that you can’t modify the FEM model and repeat the measurement but, if the model has some parts unrealistic, have to be declared specifically in the manuscript.
Some answers given to the reviewers have to be reported in the manuscript.
You described very accurately the general limits of the FEM but not the specific limitations due to the model design.
This is very important because other authors basing on your study can reproduce the same study with a model overcoming these limitations.
The most important is the absence of biological space and the crown shape. See the previous revisions.
I can see in the figures a continuous stress distribution from the abutment to the cortical bone but I think, It’s not so obvious that introducing a space of 3 mm nothing changes.
About the crown, It was declared ( to justify the particular shape), that it represents a provisional crown but the E module used for the crown is those of the ceramic. I think it is unreal because the provisional is resin-made.
For the same reason also the thickness used is better has to be notified.
The last sentence of the results “These results are in fully agreement with the observations of clinical practice.” Have to be removed because It isn’t right for the section and It’s the same as the first sentence of the discussion.
Finally, in the discussion “….very accurate modeling…..” and “……. a very realistically way……” are too strong terms for the reason described above

Author Response

I agree with the complexity of the subject but the model shown, as previously indicated, has some specific limitations which have to be clearly described. 
I can understand that you can’t modify the FEM model and repeat the measurement but, if the model has some parts unrealistic, have to be declared specifically in the manuscript.
Some answers given to the reviewers have to be reported in the manuscript.

Indeed. We reported that in the manuscript.

You described very accurately the general limits of the FEM but not the specific limitations due to the model design. 
This is very important because other authors basing on your study can reproduce the same study with a model overcoming these limitations. 
The most important is the absence of biological space and the crown shape. See the previous revisions.

Very true. We partially remedied that in the first revision and now we complete it.

I can see in the figures a continuous stress distribution from the abutment to the cortical bone but I think, It’s not so obvious that introducing a space of 3 mm nothing changes.

Introducing in the model that space of 3 mm means a different geometrical model which possibly can be studied in the future. In the case of the present model, we will assume this limitation, which will possibly be studied further.

About the crown, It was declared ( to justify the particular shape), that it represents a provisional crown but the E module used for the crown is those of the ceramic. I think it is unreal because the provisional is resin-made.

Very true, it is really a little forced. But, if there is no question of cost, and given that the mechanical properties of ceramics are clearly superior, this material is not outworn over time, referring to the masticatory surface (the resin from which the temporary crown is usually made, becomes abrasive easily and in a short period of time), this solution can be accepted without problems. Moreover, by using ceramics, which provide the brightness and translucency of the natural tooth, the aesthetic part is clearly superior, and if the patient, even if for a short period has maximum aesthetic requirements, can use this solution.

For the same reason also the thickness used is better has to be notified.
The last sentence of the results “These results are in fully agreement with the observations of clinical practice.” Have to be removed because It isn’t right for the section and It’s the same as the first sentence of the discussion.

Done.

Finally, in the discussion “….very accurate modeling…..” and “……. a very realistically way……” are too strong terms for the reason described above

Done.

Submission Date

08 July 2020

Date of this review

27 Jul 2020 22:00:34

Reviewer 3 Report

The authors do not address any of the major concerns raised. The manuscript has undergone only very minor amendments.

While I accept that my personal point of view can be challenged, the concerns raised should be addressed in the text of the paper, not only in a private rebuttal to the reviewer.

I reiterate.

Discuss how the area of study, subjected to arbitrary boundary conditions, was determined. Please note that the whole cortical bone in this area is modelled as damaged. The extent of the damaged area, and the boundary conditions defined on it should be thoroughly analysed and confronted with clinical practice. I cannot believe that the extent of the damaged cortical bone is irrelevant with respect to the results. The fact that stresses are vary small near the boundary (especially if the von Mises equivalent stress is considered) could be a byproduct of the way in which boundary conditions are imposed.

Discuss why the von Mises stress is an appropriate representation of the stress state of cortical bone. The authors state that "The use of von Mises theory can be questionable (even inappropriate) only in the case of uniform triaxial states of compression, for which the surface of the limit states is a closed surface." I do not agree: von Mises stress is appropriate (by definition) only for ductile metals; if used for different materials, compelling evidence should be presented. Please give a reference in which the use of von Mises stress is considered appropriate for assessing damage/fracture of cortical bone. In BIOMECHANICS OF TRABECULAR BONE, cit. (page 318), it is stated that even the Tsai-Wu criterium could be considered insufficient:

"In addition, theoretical formu- lations have been proposed (80, 81) based on the Tsai-Wu quadratic theory, which was originally developed for engineering composite materials (82, 83), and on cellular solid theory (79). Collectively, these studies have shown that the Tsai-Wu theory, although a good choice for axial-shear loading, does not work well for triax- ial loading because indications are that the failure envelope for the latter does not fit the ellipsoidal shape obtained from the quadratic formulation of the theory (80)."

If the authors insists on representing results by the von Mises stress, there should be a strong discussion of its validity (and please, in the paper, and not only in the rebuttal). I still think that principal stresses/strains could give a better understanding of what is going on in the cortical bone. By the way, presenting FOS values, without defining the limit stress is meaningless: FOS is (limit stress)/(von Mises stress) so this figure is not well defined, if the limit stress is not explicitly stated.

Author Response

The authors do not address any of the major concerns raised. The manuscript has undergone only very minor amendments.

While I accept that my personal point of view can be challenged, the concerns raised should be addressed in the text of the paper, not only in a private rebuttal to the reviewer.

I reiterate.

Discuss how the area of study, subjected to arbitrary boundary conditions, was determined.

The problem domain is small, but enough to have a correctly defined problem. The domain is chosen correctly, as long as the interaction of the neighborhood does not exist, it is proved by the results and can be observed in the figures. In accordance with the boundary conditions, our model is correct. If it were not correct, there would have been reactions in various areas of the field. The stress pattern in the case of our model is consistent with studies of similar boundary problems and the observations from clinical cases.

The fact that we obtained almost zero stress at boundaries shows the correctness of the choice of the domain, as well as of the boundary conditions.

The influence of implant surgery is small and does not affect large areas of bone.

We didn’t understand which is precisely the wrong or questionable result in your opinion, consequence of a wrong domain and boundary conditions choice. We add, as an example the stress pattern obtained by a quite similar study from Materials, MDPI, 2019:

Please note that the whole cortical bone in this area is modelled as damaged. The extent of the damaged area, and the boundary conditions defined on it should be thoroughly analysed and confronted with clinical practice. I cannot believe that the extent of the damaged cortical bone is irrelevant with respect to the results. The fact that stresses are vary small near the boundary (especially if the von Mises equivalent stress is considered) could be a byproduct of the way in which boundary conditions are imposed.

It is true, we have to admit that this is a limitation of the study, due to the fact that we adopted a simpler geometry model (the damaged area from the cortical bone would be estimated randomly as well). We will discuss that in the paper, we kindly thank you.

Discuss why the von Mises stress is an appropriate representation of the stress state of cortical bone. The authors state that "The use of von Mises theory can be questionable (even inappropriate) only in the case of uniform triaxial states of compression, for which the surface of the limit states is a closed surface." I do not agree: von Mises stress is appropriate (by definition) only for ductile metals; if used for different materials, compelling evidence should be presented. Please give a reference in which the use of von Mises stress is considered appropriate for assessing damage/fracture of cortical bone. In BIOMECHANICS OF TRABECULAR BONE, cit. (page 318), it is stated that even the Tsai-Wu criterium could be considered insufficient:

"In addition, theoretical formu- lations have been proposed (80, 81) based on the Tsai-Wu quadratic theory, which was originally developed for engineering composite materials (82, 83), and on cellular solid theory (79). Collectively, these studies have shown that the Tsai-Wu theory, although a good choice for axial-shear loading, does not work well for triax- ial loading because indications are that the failure envelope for the latter does not fit the ellipsoidal shape obtained from the quadratic formulation of the theory (80)." If the authors insists on representing results by the von Mises stress, there should be a strong discussion of its validity (and please, in the paper, and not only in the rebuttal).

We definitely could not say that von Mises stress is an appropriate representation, but it is rather a convenient one, that is largely used even in the bone or ceramic materials. As the trabecular bone possesses weaker mechanical properties, it plays a secondary role in approaching the progress of osseointegration from a biomechanical point of view. We will also discuss that in the paper, we kindly thank you.

I still think that principal stresses/strains could give a better understanding of what is going on in the cortical bone. By the way, presenting FOS values, without defining the limit stress is meaningless: FOS is (limit stress)/(von Mises stress) so this figure is not well defined, if the limit stress is not explicitly stated.

Yes, we will discuss more the FOS values.

Our model was thought to focus on osseointegration progress. The complexity of the osseointegration process means that the modeling cannot cover the whole multitude of aspects involved.

Any numerical analysis of some structures, especially complex ones (by geometry and materials), is recommended to be studied first of all in static regime, elastic and isotropic material, and generally in the most simple model possible, but taking into account the main properties according to the study targets. If it provides necessary and useful data for the proposed purpose, the analysis may be sufficient. Agree that no research is completely completed; often new aspects appear to be researched, but the researcher decides for a work if his numerical investigation continues or if the one carried out meets the declared purpose of the work.

The model presented in the article represents an approximation of the process and which, for the moment, does not take into account all the aspects related to osseointegration, but which can be, undoubtedly, the object of future calculations, in light of the topics you have brought up. Thank you for your valuable observations.

Submission Date

08 July 2020

Date of this review

26 Jul 2020 22:58:00
